# A New Framework for Smart Doors Using mmWave Radar and Camera-Based Face Detection and Recognition Techniques

**DOI:** 10.3390/s24010172

**Published:** 2023-12-28

**Authors:** Younes Akbari, Abdulaziz Al-Binali, Ali Al-Mohannadi, Nawaf Al-Hemaidi, Omar Elharrouss, Somaya Al-Maadeed

**Affiliations:** Department of Computer Science and Engineering, Qatar University, Doha P.O. Box 2713, Qatar; aa1802173@student.qu.edu.qa (A.A.-B.); aa1807781@student.qu.edu.qa (A.A.-M.); na1912009@student.qu.edu.qa (N.A.-H.); elharrouss.omar@gmail.com (O.E.); s_alali@qu.edu.qa (S.A.-M.)

**Keywords:** smart doors, mmWave radar, camera, face detection, face recognition

## Abstract

By integrating IoT technology, smart door locks can provide greater convenience, security, and remote access. This paper presents a novel framework for smart doors that combines face detection and recognition techniques based on mmWave radar and camera sensors. The proposed framework aims to improve the accuracy and some security aspects arising from some limitations of the camera, such as overlapping and lighting conditions. By integrating mmWave radar and camera-based face detection and recognition algorithms, the system can accurately detect and identify people approaching the door, providing seamless and secure access. This framework includes four key components: person detection based on mmWave radar, camera preparation and integration, person identification, and door lock control. The experiments show that the framework can be useful for a smart home.

## 1. Introduction

Home security is indeed one of the crucial categories in a smart home [1]. In addition, the COVID-19 pandemic has fundamentally changed our lives, and the demand for non-contact or touch-free options has increased to reduce the risk of transmission through surfaces. While smart door security systems are available on the market, they often require users to initiate the authentication process by clicking a button, which may not be desirable for some individuals due to concerns about hygiene or the risk of transmission. Research categories related to smart doors are classified into four groups: manual video identification, intrusive biometric identification, non-intrusive automatic visual identification, and non-intrusive automatic audio identification. Systems based on facial recognition (non-intrusive automatic visual identification) are growing in these groups. It was shown in [2] that face recognition is indeed a viable method for smart doors and has gained significant popularity in recent years. The global market for facial recognition door locks is expected to reach a value of approximately USD 738 million in 2020. However, the market has faced challenges due to manufacturing restrictions brought about by the COVID-19 pandemic, particularly in the residential sector, which accounts for 32.4% of demand. Meanwhile, the commercial and hospitality sectors make up approximately 45% of market demand, with the remainder split between the government and other sectors. In North America, the market is estimated to be worth USD 352 million in 2020 and is expected to reach USD 1 billion by 2028 [3]. Some of the key players in this market include Samsung Group, ZKTeco, and WISEPOCH, all of which offer facial recognition door lock products ranging in price from USD 630 and up.

This work focuses on three main parts for the development of a smart door: person detection based on camera and mmWave radar, integration of the results of the sensors, and face detection and recognition. Although traditional methods have achieved promising results in recognizing people based on images and videos, deep learning methods have been increasingly used in recent years [4]. Person detection based on mmWave has also received a lot of attention in recent years and the industry has improved sensors with high accuracy to detect multiple people [5]. In face detection and recognition, there are improvements based on traditional and deep learning approaches [6,7]. There is a lot of work being carried out on smart doors based on facial recognition, some of which has recently been implemented to increase accuracy. However, only limited papers on smart doors based on facial recognition are peer-reviewed in reputable journals and conferences. In [8], an efficient attitude-tracking algorithm (EATA) for smart doors based on face recognition is presented. In the paper, a software-based analysis was performed without using any hardware. A smart door system named ESP32-CAM is presented in [9]. Facial recognition was considered in the development of a highly reliable door lock system. The system performs real-time face recognition and quickly detects when a face is in front of a camera. The face recognition system was developed based on machine learning, with a Haar graphics recognition program built into OpenCV in [10]. Also, the system used deep learning methods such as VGG-16, VGG-19, and ResNet50. In [11], a system has been developed that allows access by simply capturing facial snapshots using a Convolutional Neural Network (CNN) approach for face recognition. The implementation was realized on a Raspberry Pi and a Pi camera. A system using a modified ResNet-50 model working as a unit was presented in [12] to provide a contactless solution based on face recognition. The deep learning method is implemented on a Raspberry Pi.

Although the works reported good accuracy, the systems used only cameras for the detection process and there are some limitations in using only cameras. When people overlap, parts of people may be covered or obscured by others. As a result, the camera may only capture partial views or distorted representations of the people, making it difficult for conventional computer vision algorithms to accurately recognize and identify each person. To overcome this limitation, some radar-based sensors such as mmWave radars can be used in detecting objects. However, radars have limitations in the visualization and detection of objects. For example, in a person detection problem, when the persons overlap together, the results report low accuracy versus other sensors such as mmWave radars. For example, in Figure 1, overlapping people are shown in which the results of the sensors are not equal. MMwave Radar detects three people whereas the camera detects two people. Also, using a mmWave sensor can notify the system of the presence of a person to start the facial recognition process. Further details on existing methods for integrating mmWave radar and camera can be found in [13,14,15]. The review papers present all the challenges associated with integration, such as synchronization and tracking multiple people.

To address these challenges, this paper presents a four-component framework that combines mmWave-based person detection, camera preparation, person identification, and door lock control for smart doors, as shown in Figure 2.

The mmWave radar technology employs high-frequency electromagnetic waves to detect and track objects within its field of view. In the proposed framework, mmWave radar is utilized to detect the presence of individuals approaching the smart door. It provides real-time information about the person’s position, velocity, and distance, enabling accurate and reliable detection even in challenging environmental conditions. To capture visual information for a person identification step, camera preparation is an essential component of the framework. Prior to person detection, the cameras are positioned and calibrated to cover the area of interest, ensuring optimal coverage for facial recognition tasks. This step involves camera alignment, focus adjustment, and other necessary preparations to maximize the accuracy and reliability of subsequent person identification. Person identification is a crucial step in the access control process. In the proposed framework, after the person is detected using mmWave radar and camera, the camera captures the person’s face for identification. Computer vision algorithms are applied to extract facial features, which are then compared with a pre-existing database of authorized individuals. This process enables accurate identification and authentication, enhancing the security of the smart door system. In this step, adding other methods such as voice recognition and verification is possible. Once the person has been successfully identified and authenticated, the door lock control component ensures seamless access to the secured area. This component interfaces with the locking mechanism of the smart door, enabling the door to be unlocked automatically for authorized individuals. Additionally, it can log access events for security monitoring and management purposes.

The four-component framework offers several benefits, including increased accuracy in person detection and identification, robustness against environmental factors, and enhanced security against unauthorized access. It can be applied in various scenarios such as residential buildings, commercial spaces, and public facilities, providing a reliable and efficient access control solution. As mentioned earlier, one of the biggest challenges during the pandemic has been to minimize physical contact with surfaces. The proposed system enables touchless access control, allowing people to enter or exit the building without physical interaction with key cards, handles, or biometric scanners. Identifying people by their faces can solve this challenge and is also a secure solution. Experiments were conducted using real-world scenarios to evaluate the framework’s performance. The results demonstrate the effectiveness of the mmWave-based person detection, camera preparation, person identification, and door lock control components, showcasing their integration and overall system performance.

As a result, the work is based on some exciting devices and methods that have been combined and adapted to develop a new way for the smart door. To the best of our knowledge, this is a new framework for using two sensors for this problem. Below is a summary of the main contributions of this paper:
A novel smart door framework that combines facial recognition and detection techniques based on mmWave radar and camera sensors.By integrating mmWave radar and camera-based face recognition algorithms, the system can accurately detect and identify people approaching the door and overcome the limitations of using only one camera, such as overlapping and detection in low-light conditions.

## 2. Proposed Framework

Four components of our framework are explained in the following subsections.

### 2.1. Person Detection

Since most of the systems are based on real-time methods, accuracy plays an important role in marketing; presenting an accurate product has a competitive advantage in the real world. Exploring people’s counting, activities, characteristics, and health issues from recoding continuously and consistently, human monitoring has provided an attractive area for research. Also, presenting an accurate system in applications such as surveillance needs specific attention. Sensors can play the role and assess in order to obtain useful information. Among the devices, the camera provides a series of visual data around illumination and distance limitations. Also, obtaining accurate results depends on the conditions and situations in which the camera is used, e.g., overlapping people, as shown in Figure 1. On the other hand, to overcome the limitation, some sensors based on radar such as millimeter-wave radars can be used alone or combined with other sensors, and obtain good results in detecting objects. However, the radars have limitations in visualizing and recognizing them. Integrating machine vision using a camera with millimeter-wave radar can make a breakthrough in all automatic applications. The accuracy of detection using mmWave can be helpful when combined with the camera for computer vision tasks in terms of visualization using a camera and rebuses of detection using radar.

To the best of our knowledge, more products presented for this purpose are solely based on a camera controlled by software. To obtain more accuracy in detecting persons, we use mmWave radar. The radar to detect objects uses the frequency-modulated continuous wave (FMCW) technique. The technique uses frequency modulating on a continuous signal to obtain velocity and range information from a radar. In the component, in our design, we have utilized a Raspberry Pi wide camera (Waveshare, Shenzen, China) that was mounted on a Raspberry Pi 3 B+ [16], a TI short-range radar evaluation board (IWR1642) [17], and a power bank. This is our device that is presented in [18], as shown in Figure 3e.

We include the TI short-range radar system in our system. The TI IWR1642BOOST evaluation board, as shown in Figure 3a, is a short-range radar system based on the IWR1642 radar chip. In the IWR1642, FMCW radar technology is used by single-chip mmWave sensors, which can be for self-monitored, low-power, ultra-accurate radar systems in industrial areas and various applications. To achieve the best performance, some features of the sensors should be noticed such as the field-of-view (FOV). The Raspberry Pi camera has a larger field of view (160 degrees, while other normal cameras usually have 72 degrees). The wide camera mounted on a Raspberry Pi has a resolution of 1920×1080 pixels, and the horizontal and the diagonal FOV are 132 deg and 160 deg, respectively. In addition, the IWR1642 horizontal FOV is 120 deg with an angular resolution of 15 deg.

The procedure for detecting and tracking persons using the IWR1642 sensor is shown in Figure 4. Person detection is implemented as a digital signal processing (DSP) code that runs on the C674x core in the IWR1642. The tracking code is executed on the ARM^®^ Cortex^®^-R4F processor (ARM Holdings plc, Cambridge, UK). The mmWave radar technology is designed to detect and track multiple moving objects within its field of view. Sophisticated signal processing algorithms from the sensor manufacturer analyze the radar returns to distinguish between different targets based on their movement patterns, speed, direction, and size. These algorithms allow the system to track multiple people simultaneously and monitor their positions in real time. In our case, we use the IWR1642 radar from the Texas Instrument (TI) family. This sensor uses micro-Doppler, which generally refers to the Doppler information generated by movements of individual parts of a particular target. The micro-Doppler features can be used to determine the characteristics of multiple targets for tracking and detection [13,19]. The steps are summarized as follows:

RF front end: Antenna as a receiver with the components processes gauged data (signal) at the original incoming radio frequency (RF), and then analog to digital converter (ADC) converts it to a digital signal.Range Processing: The step processes the signal frames by the active chirp time. For each antenna, 1D fast Fourier transform (FFT), and 1D windowing should be applied.Capon beam forming: An algorithm that integrates inverse-angle spectrum generation, co-variance matrix generation, and static clutter removal with a range-angle heat map as the output.Constant false alarm rate (CFAR) detection method: It consists of a constant false-alarm rate and two-pass. To find detected points, after obtaining the smallest CFAR-CASO in the range domain, it should be confirmed by the second smallest CFAR-CASO.Estimating Doppler: Capon beam-weights of the detection module obtain a detected (range, azimuth) pair and follow the Doppler estimation by filtering the range bin. Finally run a peak search over the FFT of the filtered range bin.Tracking step: Obtaining updated and localization points and submitting them for a new decision. The step is dedicated to obtaining a set of trackable objects with point density, physical dimensions, and certain properties like position and velocity.

### 2.2. Camera Preparation and Integration

In the component, we use the camera to detect people in order to have a comparison with the results of the mmWave radar. The integration of the results is then used for the face recognition component.

There are many methods for recognizing people with the camera, such as those in [20,21]. Among others, deep learning methods have achieved good results in practice. However, to run a deep learning method on the Raspberry Pi 3 B+, we have to accept some limitations, such as the need for more power to process the methods. Therefore, we use a simple, accurate method, which is compatible with the state-of-the-art method to detect objects. Because the method presented in [22] decreases the computational cost considerably, we select it to detect persons using video captured by the camera. This method is based on the use of fast feature pyramids with a distinct detection framework including aggregate channel features (a novel variant of integral channel features). In the aggregate channel features (ACF) method [23], several channels are considered as *C* = (*I*) where *I* is an input image. Then, to smooth the resulting lower resolution channels, it computes the number of pixels in every block. The method considers single-pixel lookups in the aggregated channels as features. To classify background from detected objects, decision trees are trained over these features. The overview of the detector is shown in Figure 5. It should be noted that in downsampling the image to extract the feature, a fast feature pyramid is considered.

In the next step, we use an integration method of the two sensors to achieve a reliable result, as shown in Figure 6. Integration methods can be categorized into three levels [24,25], which are before object detection (bottom level) [26,27], during object detection [28,29], and at the state level [30,31]. In our system, the integration step is performed at the object-detection level, which provides a various list of stationary and moving objects around both the camera and mmWave sensor environment to detect moving objects simultaneously. The integration of mmWave radar and camera sensors for people detection brings several considerations and challenges, mainly related to data synchronization, sensor fusion, and the alignment of information from these different sources. To integrate the two sensors, a global sampling time is defined, and both sensors report their results based on the current frames in the time. In our experiments, the mmWave radar has a delay of 20 frames to detect people when it starts. After the people counting by the sensors, if the number of people is equal based on the two sensors, the face detection and recognition component is called. Otherwise, this process continues until the number of people is equal. In our implementation, after matching the results of the two current frames, the current frame in the camera is used to detect and recognize the face. Depending on the security level, the number of frames can be increased. The nature of the application, that persons stay in front of the doors for a while, may lead us to ignore complex synchronization for the sensors.

To visualize the relationship between the two sensors, we also mapped radar coordinates to camera coordinates in the component. In general, the mapping cannot improve the overall accuracy of the system. This mapping only helps to check whether or not both sensors are at the same position with the same number of people. This process (Figure 7) is only necessary for the first time. After finding the mapping process, we do not need to repeat it. After relating the two coordinates, two sets of points on the common plane are obtained. To obtain a common position for the two sensors, we fused the detected objects’ positions, and in our implementation, the two coordinates were related to the camera’s coordinates, as shown in Figure 7.

The Kalman filtering technique [32] was used to fuse the objects’ positions, which is accurate in estimating a global position. This method has been used in several successful works for this purpose [15]. Since the detected object position on the sensor is more accurate than the camera, we related the two coordinates to obtain an accurate result. A global coordinate system should be defined for multiple sensor systems because each sensor has its coordinate system. For the camera, we use the standard pinhole model. Concerning the camera, we have the image plane coordinate (u,v), the camera coordinate $\left(x_{c},y_{c},z_{c}\right)$, principal point coordinates $f_{x},f_{y},c_{x}$, the x,y direction focus length $c_{y}$, and matrix of intrinsic parameters *K*.
(1)$$z_{c}\left[\begin{array}{c} u\\ v\\ 1 \end{array}\right]=K\left[\begin{array}{c} x_{c}\\ y_{c}\\ z_{c} \end{array}\right]=\left[\begin{array}{ccc} f_{x} & 0 & c_{x}\\ 0 & f_{y} & c_{y}\\ 0 & 0 & 1 \end{array}\right]\left[\begin{array}{c} x_{c}\\ y_{c}\\ z_{c} \end{array}\right]$$

As shown in Figure 7, $\left(x_{r},y_{r},z_{r}\right)$ represents the position of an object based on radar. The radar provides only two coordinates and information on a 2D plane. The matrix M^RT^ (including the rotation R and translation T) obtains a relationship between the camera coordinate $\left(x_{c},y_{c},z_{c}\right)$ and the radar coordinate $\left(x_{r},z_{r}\right)$:(2)$$\left[\begin{array}{c} x_{c}\\ y_{c}\\ z_{c} \end{array}\right]=M^{RT}\left[\begin{array}{c} x_{r}\\ y_{r}\\ 1 \end{array}\right]=\left[\begin{array}{ccc} m_{11} & m_{12} & m_{13}\\ m_{21} & m_{22} & m_{23}\\ m_{31} & m_{32} & m_{33} \end{array}\right]\left[\begin{array}{c} x_{r}\\ z_{r}\\ 1 \end{array}\right]$$

As mentioned above, we have two sets of points on the common plane, which we consider all points after mapping on camera coordinates. The main purpose of the data fusion is that while we obtain uncertainty object positions by multi-sensors separately, as a rule of thumb, accurate points can be found and combined by multi-sensors in a common coordinate. As a result, we can have more reliable positions compared to the individual sensor, and even data captured by the multi-sensor without fusing them avoid degrading the quality of measurements [33].

### 2.3. Person Identification and Door Lock Control

Since the two components are very closely related, they are explained in a subsection. The devices used in the components are a light-dependent resistor (LDR), LED, Buzzer, Solenoid Lock, and SMS service. The structure of the two components is shown in Figure 8. The LDR uses a light sensor that would turn the LED on and off depending on the ambient light surrounding the door. In a real situation, we need enough light to detect and recognize faces. In the component, when a person is detected as explained in the previous section, the face is now detected and the image is cropped to the region of the face. This cropped image is used to identify whether the person is allowed in with the face recognition algorithm. The face recognition algorithm is a trained model that the model trains a selected set of people that are “allowed in”. The way training is by recording these people’s faces from all sides and using these recordings to train our model. For the training step, we train our model on recognizing a person given a specific name. As mentioned in the previous subsection, there are some limitations to running some deep learning methods on the type of Raspberry Pi. The Histogram of Oriented Gradients (HOG) feature extraction method [34,35] and a support vector machine (SVM) classifier [36] are used for face detection and recognition. The HOG is a powerful technique for extracting features from images, especially for object-recognition tasks. It captures information about local object shapes and structures by analyzing the distribution of intensity gradients or edge directions in an image. HOG descriptors are insensitive to changes in lighting conditions and are therefore suitable for scenarios where illumination variations can negatively affect conventional feature extraction methods. HOG effectively encodes the texture and edge information within local regions of an image, providing discriminative features that can be critical for distinguishing between facial features such as the edges of the eyes, nose, and mouth. The feature vectors extracted by HOG often represent high-dimensional data. The ability of a SVM to process high-dimensional feature spaces without overfitting makes it an excellent choice for classification tasks such as face recognition. The combination of HOG for feature extraction and a SVM for classification in face recognition utilizes the discriminative power of HOG descriptors to robustly represent facial features. These descriptors, when fed into a SVM classifier, enable efficient and accurate classification of facial images into relevant classes.

In the component, we use the handle to open and close the door. Once a person is recognized, the door lock will open, allowing the person in. It will lock again once the person is out of the frame of the camera and an SMS message is sent to an administrator letting them know that the recognized person got inside the door. If an unrecognized person was present in front of the door. The system will alert a buzzer until the person is out of the frame.

The devices of the component are connected to the Raspberry Pi, as shown in Figure 9 (the connections for each part are marked with a specific color). We connected our Solenoid Lock to an external 12 V power supply, this is because the Solenoid Lock requires a 12 V power supply. Because the Raspberry Pi’s GIPOs can run on 5 V at its max, we needed a 5–12 V relay to bridge the gap between the Solenoid Lock and the Raspberry Pi and to be able to control the lock. The circuit also includes a LDR (Light-Dependent Resistor) that uses a light sensor that turns the LED on and off depending on the ambient light surrounding the door. Based on the documentation, the LDR needs a 0.1 μf capacitor to stabilize the readings. We used a buzzer to sound an alert once an unrecognized person was at the door. The buzzer’s connection is very simple: it is connected directly to the Raspberry Pi since it runs on 5 V also.

## 3. Evaluation

To evaluate the whole system, we evaluate the two main components which are the detection process by mmWave and camera and the door lock process. These experiments were conducted in Qatar University laboratories and for evaluating mmWave, eight people (two female and six male) participated in them. We look at 20 frames to decide that both the sensor and the camera see the people to be processed, during the frames. The field of view of the mmWave sensor and camera is 120 horizontally. To test the effectiveness of the presented system, two sets of experiments in many parts are conducted with the portable device through indoor and outdoor conditions. In the first set, the accuracy of the fusion method is tested while in the second set, object detection results in terms of radar and camera are tested. The details of the experiments and the comparison with the state of the art are explained in these subsections.

### 3.1. Experimental Results on the First and Second Component

#### 3.1.1. Integration Method Tracking Results

We consider the experiments only in an indoor situation with tracking four persons on four different paths as shown in Figure 10a, in which the participants walked based on the paths. We performed the experiments four times. The first time, only one person walked on the paths, and for each next time, one person added to others, which means that the fourth time, all the people are on the paths simultaneously. The paths are specified by four-color labels and considered the ground truth. Also, it should be noted that however the experiments are carried out under a controlled environment, the scenarios are not known to the camera and radar. As can be seen in the figure, using the fusion method obtains more accurate trajectories than when the tracking step is conducted by only radar or camera.

#### 3.1.2. Person Detection Results

We have conducted many scenarios for the experiment. Results regarding people recognition based on the ACF method and radar FMCM are shown in Table 1. During the experiments, we obtained the results in real time and also we recorded the corresponding video, to analyze each frame in the future. The tracking evaluation process is performed respectively by the radar and camera. When the radar and the camera have the same number of detected objects, data fusion is carried out. We did some experiments in outside and inside scenes that as can be seen in the statistical results from Table 1, the mmWave radar detection results are better.

We can see that the radar detected all persons successfully in all fused scenes. It should be noted that the radar detects objects after taking some frames. When our experiments are outdoors in a low illumination situation, the camera can not detect any objects. As visualized in the results (Figure 11), the detection method by the camera can be promising in a suitable situation. While the MMW radar detection and tracking results are stable in low illumination (indoor and outdoor), the camera has a problem detecting persons. As shown in Figure 12, a scene with many objects as obstacles such as tables and chairs is selected. In the scene, in some frames, the detection based on the camera obtained some error in detecting other objects instead of the persons.

### 3.2. Experimental Results on the Third and Fourth Component

The first key functionality is the face detection. For a reliable product, our system should be able to detect faces consistently and accurately as soon as a person approaches the door. Therefore, in our first testing stage, we tested our face detection algorithm with static images. We tested the algorithm on men and women and had a 99% percent accuracy rate. Some of the test images and the results are in Figure 13. The second key functionality is face recognition. Recognizing a face in real-time with high processing speed and high accuracy was vital to our project. So for our tests, we put together a group of 15 people, of whom we decided to let the system recognize 7 and not allow the other 8. Figure 14 shows the results of our testing for three persons. Next, we started testing our LDR with the LED circuit running multiple tests by covering the LDR and then eventually turning the light in the room on and off and the results are as follows. After testing the LDR and the LED, we moved on to testing the buzzer. The test was straightforward. As soon as an unrecognized person came into frame, the buzzer would medially start buzzing and as soon as the person stepped out of frame, the buzzer would stop. Our last feature to test was sending an SMS as soon as a person was recognized and allowed in the door. We ran multiple tests, and the results were constant. The SMS was being sent to the administrator’s phone while correctly identifying the person and showing their name in the SMS message. Once we tested each use case separately; we ran many tests for the system. All the components and features were running together as tested earlier. In our testing, we concluded that our accuracy rate was 99% in real-time with no mistakes. The samples of the results are shown in Figure 15. Finally, a comparison is made with the state of the art in facial recognition based on smart door applications (see Table 2). It should be noted that the methods were evaluated on the basis of their databases. Therefore, a fair comparison with the methods is very difficult. However, our method and other methods for face recognition are successful and all results are better than 95%.

## 4. Conclusions

This paper proposes a comprehensive four-component framework for smart doors, combining mmWave-based person detection, camera preparation, person identification, and door lock control. The integration of these components provides an efficient, accurate, and secure access control system. The secure aspect of the framework is based on some limitations of the camera, such as overlapping of people and lighting conditions, which are solved by using mmWave radar. The mmWave radar can recognize people more accurately than the camera. However, we should use a camera to recognize faces. The experimental results highlight the potential of the proposed framework for various real-world applications. The proposed framework can be used with some modifications and adaptations in different applications with different security levels.

For future work, we believe that one of the main problems to be solved is to improve our processing speed without compromising our accuracy, which is a very complicated challenge. We would also like to improve our system by adding sound recognition. This would work so that the person can either use voice recognition if face recognition fails, or the system can be set up so that both face and voice recognition are required to unlock the door. With this feature, the user would have more options and more settings to configure their security standards.

## Figures and Tables

**Figure 1 sensors-24-00172-f001:**
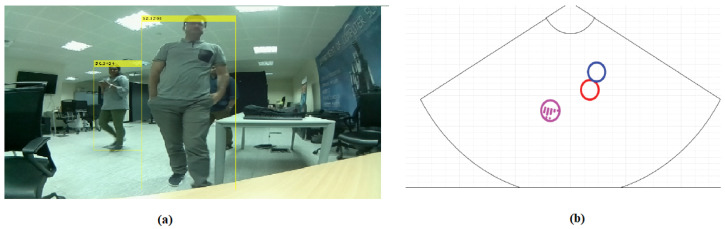
A result sample of object detection by (**a**) camera and (**b**) mmWave radar.

**Figure 2 sensors-24-00172-f002:**
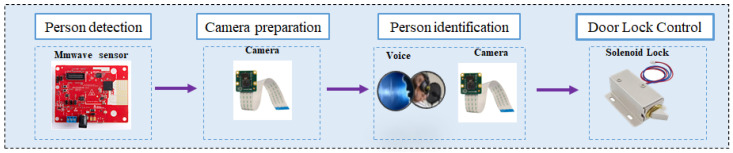
Overview of our framework.

**Figure 3 sensors-24-00172-f003:**
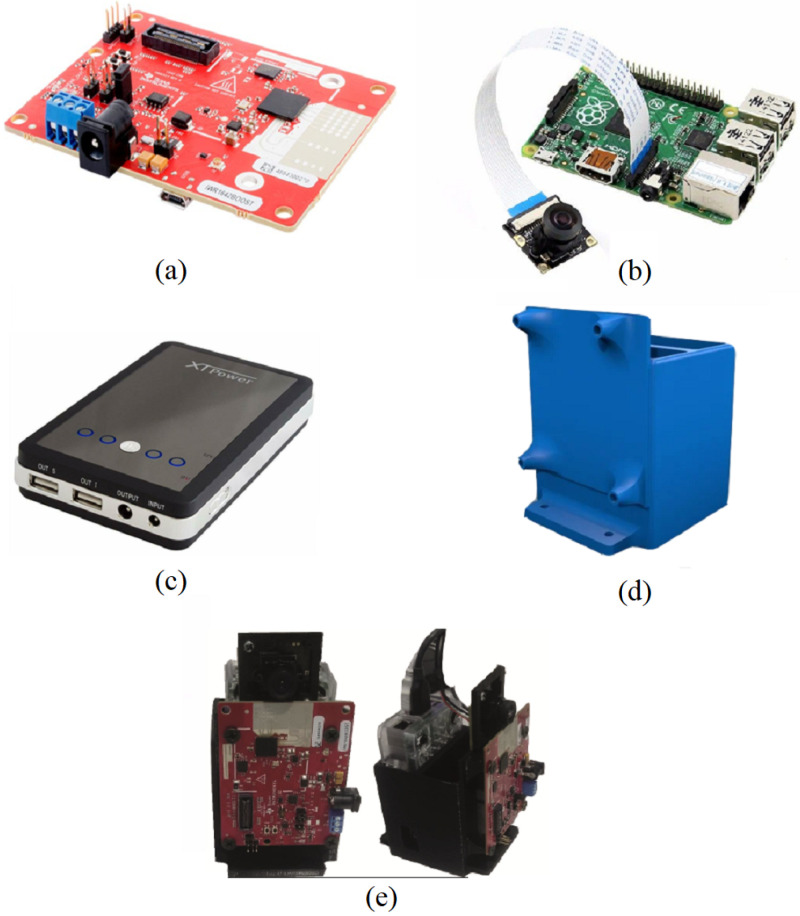
Hardware parts of the component: (**a**) TI short-range radar evaluation board (IWR1642), (**b**) Raspberry Pi 3 B+ and camera, (**c**) power bank, (**d**) case, (**e**) device assembled as presented in [18].

**Figure 4 sensors-24-00172-f004:**
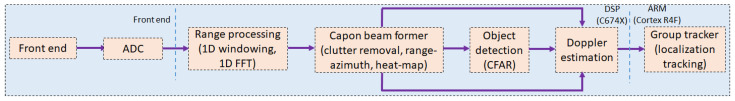
The basic steps of object detection and tracking by the radar [17].

**Figure 5 sensors-24-00172-f005:**
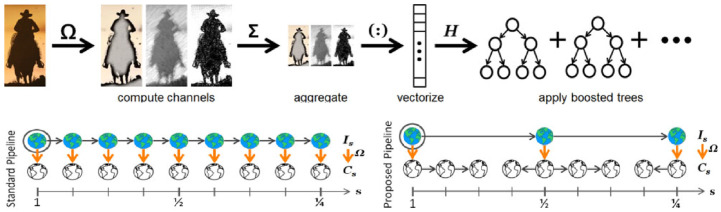
The overview of the detector ACF [23].

**Figure 6 sensors-24-00172-f006:**
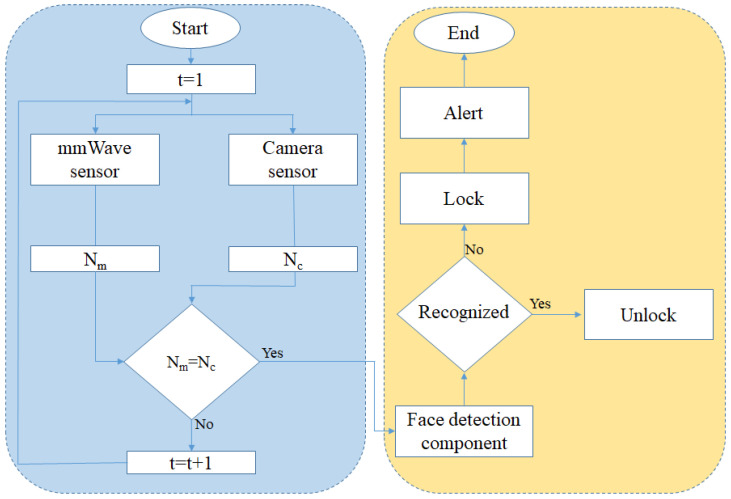
Flowchart of the integration of the camera and the mmWave sensor for two components of people detection and face recognition (the flowchart is only for one cycle of the two components. *t*, $N_{m}$, and $N_{c}$ are a global sampling time, the number of people detected by the camera, and mmWave sensor, respectively).

**Figure 7 sensors-24-00172-f007:**
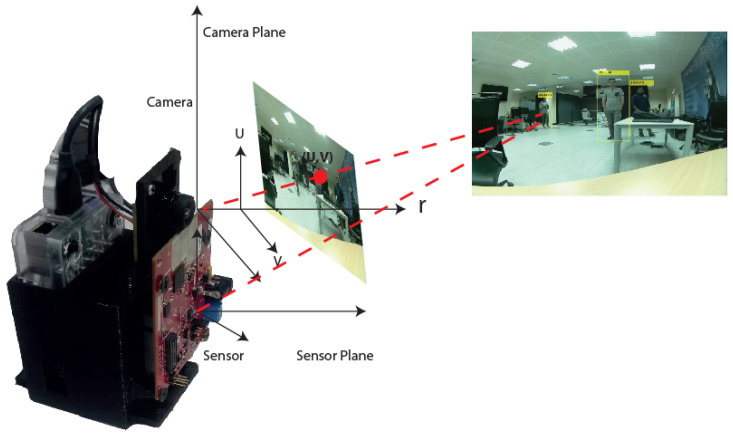
The RGB-D sensor coordinate and MMW radar coordinate.

**Figure 8 sensors-24-00172-f008:**
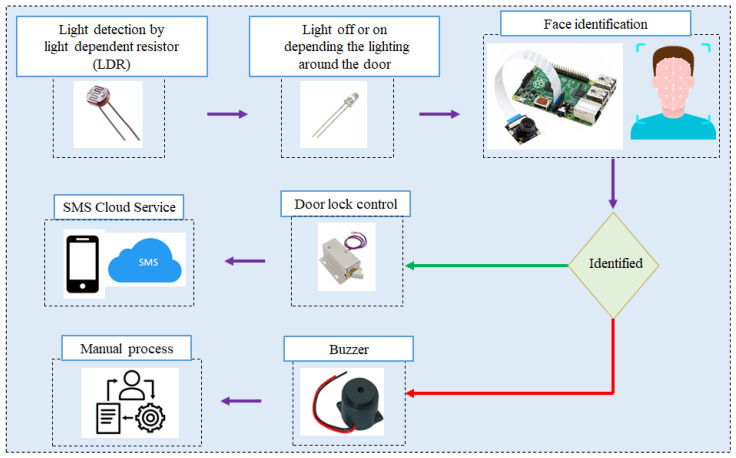
Our process for person identification and door lock control.

**Figure 9 sensors-24-00172-f009:**
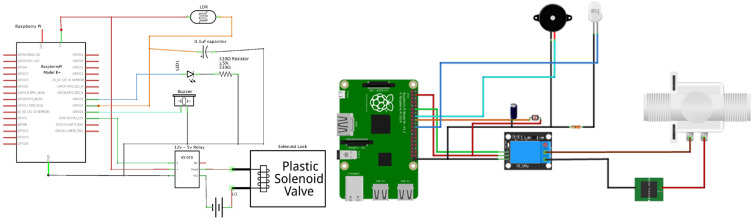
The connection between hardware components (LDR, LED, Buzzer, Solenoid Lock) and Raspberry Pi (the connections for each device are shown in a specific color).

**Figure 10 sensors-24-00172-f010:**
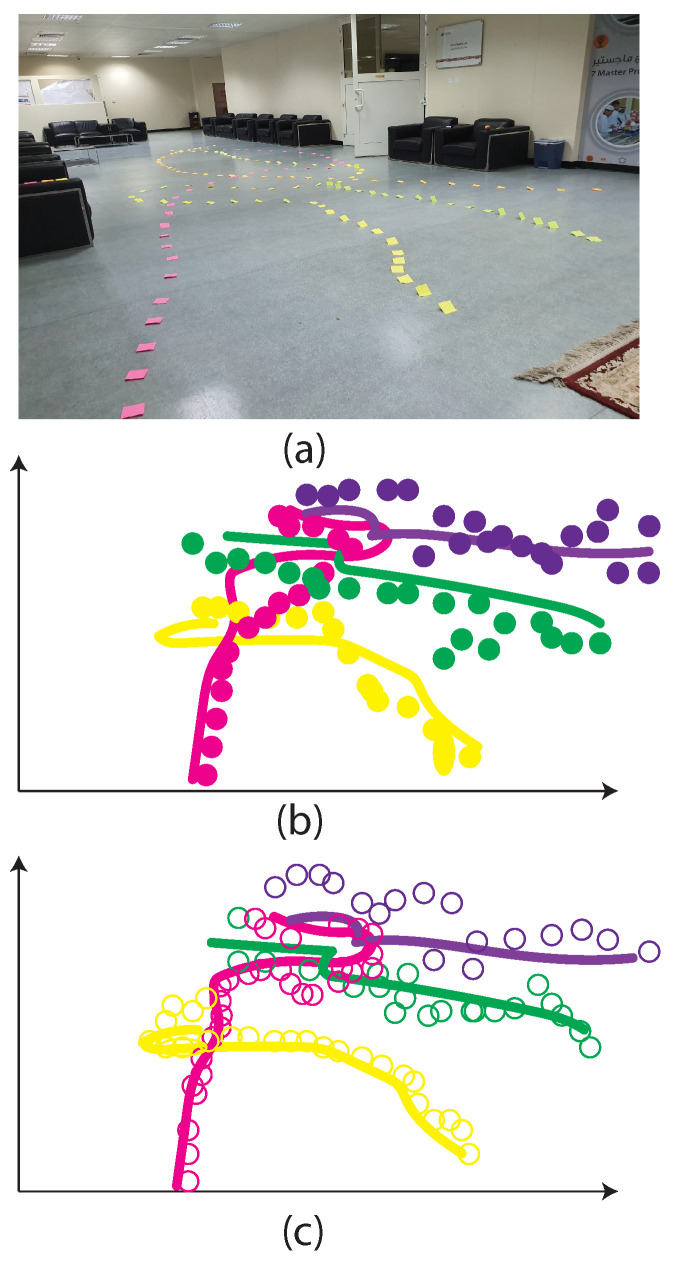
Trajectory comparison, (**a**) ground truth: red, green, yellow, and violet, estimated trajectory: circles in the same colors. (**b**) people detection by radar and (**c**) people detection by fusion of radar and camera.

**Figure 11 sensors-24-00172-f011:**
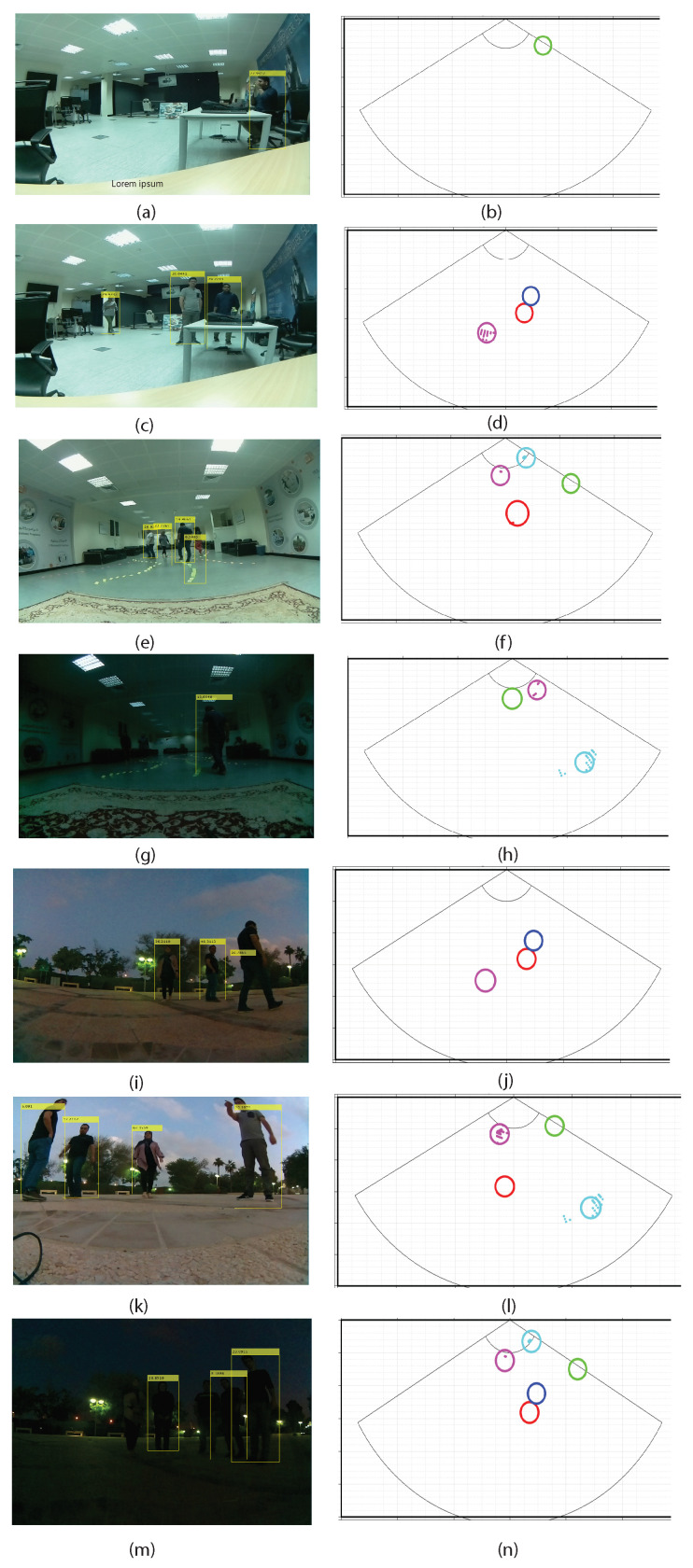
Samples are conducted indoors and outdoors. (**a**,**c**,**e**,**g**,**i**,**k**,**m**) are the results of people detection based on camera and (**b**,**d**,**f**,**h**,**j**,**l**,**n**) are the results of people detection based on mmWave radar. In (**g**,**m**), conducted indoors and outdoors in poor lighting, the camera cannot properly see the persons.

**Figure 12 sensors-24-00172-f012:**
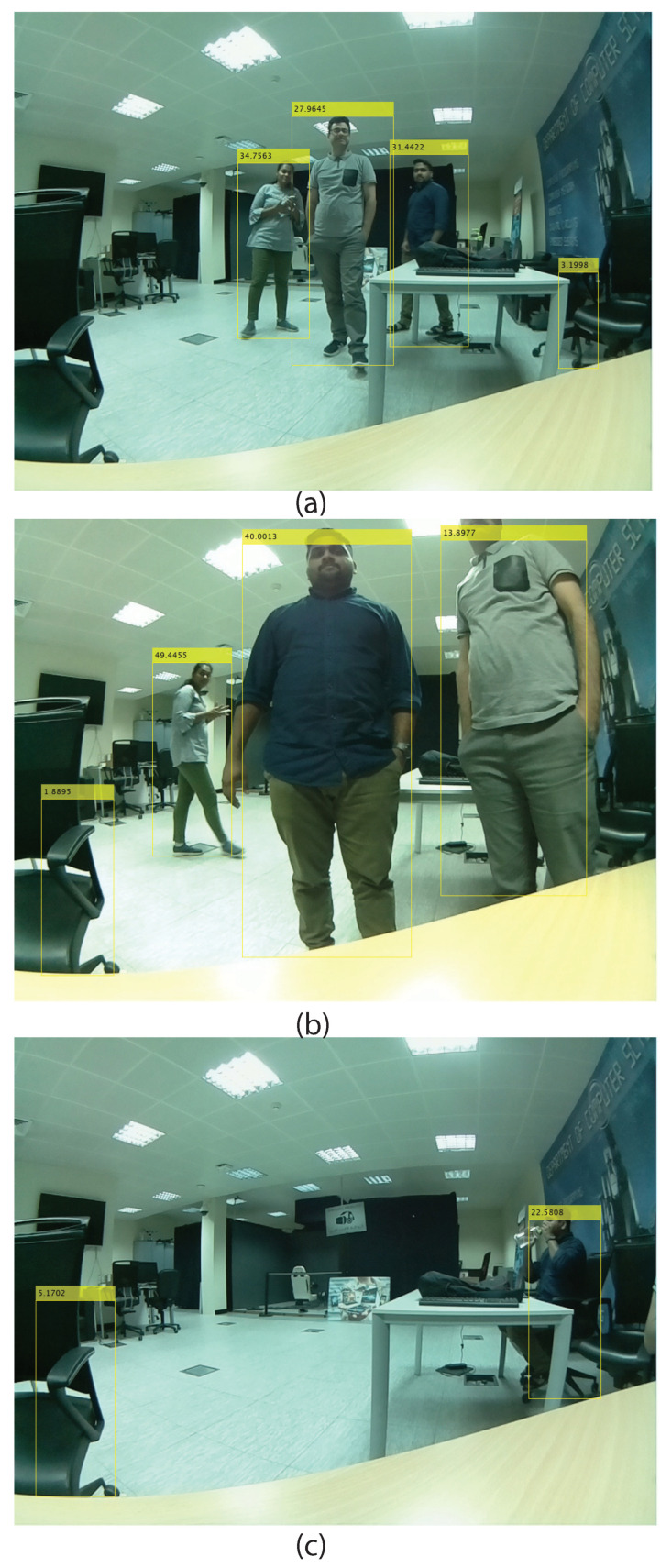
A scene with many objects as obstacles (**a**–**c**) such as tables and chairs to evaluate the person detection method.

**Figure 13 sensors-24-00172-f013:**
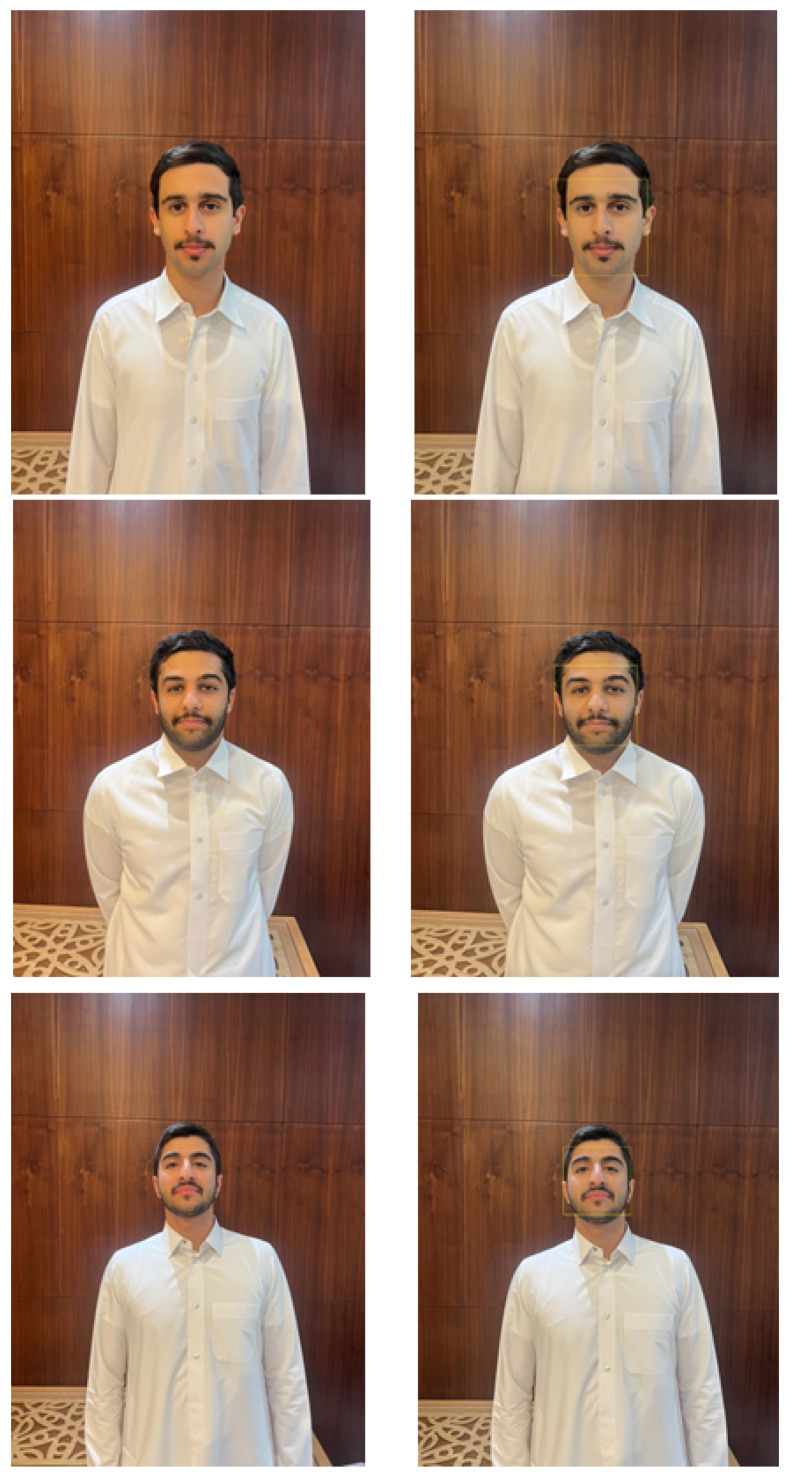
The samples of the result of the face detection method.

**Figure 14 sensors-24-00172-f014:**
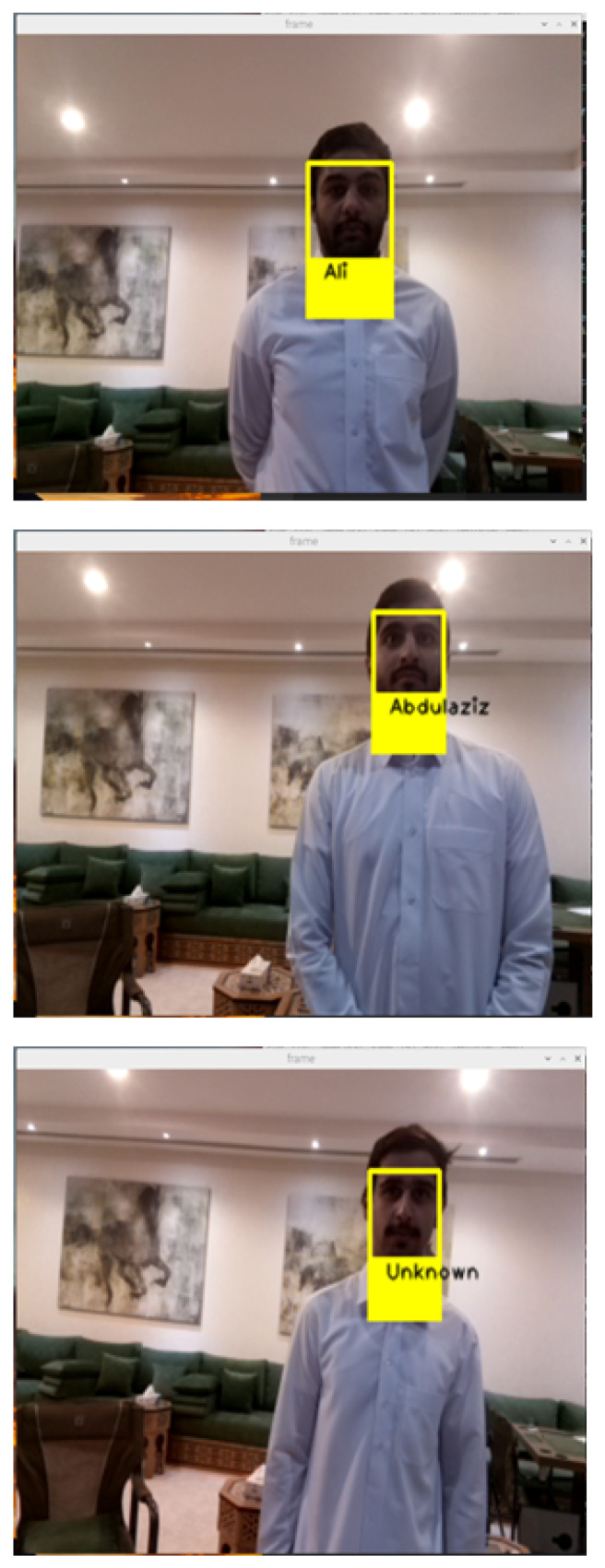
The samples of the result of the face recognition method.

**Figure 15 sensors-24-00172-f015:**
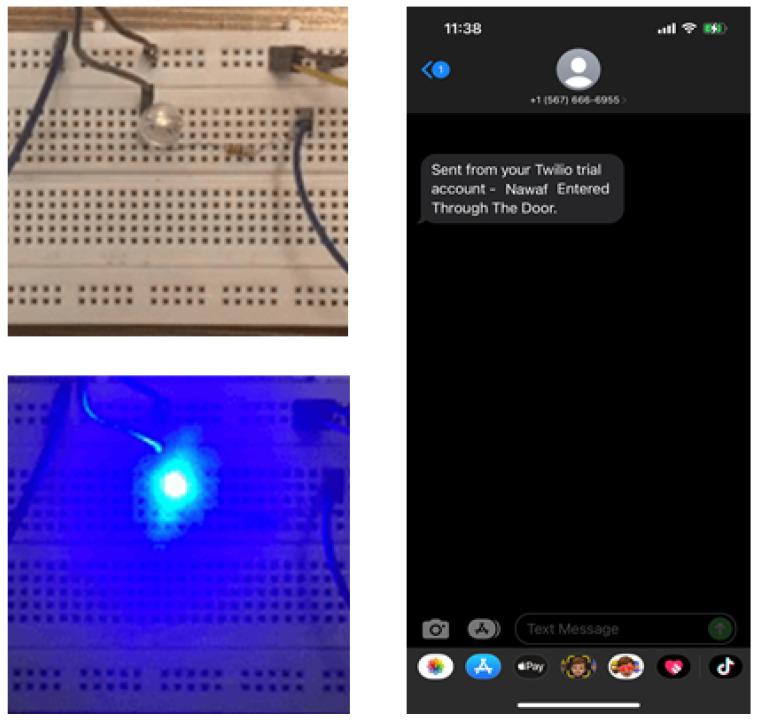
The results of light detection by LDR and a message sent to the homeowner.

**Table 1 sensors-24-00172-t001:** The results based on sensor, camera, and fusion for people detection.

485 Sensor Frames	485 Camera Frames	270 Fusion Frames
404 frames	270 frames	270 frames
83.30%	55.67%	100%

**Table 2 sensors-24-00172-t002:** Comparison of our method based on HOG feature extraction method and SVM classifier with other face recognition methods in smart door application in terms of accuracy (%).

Methods	Database	Accuracy
[11]	Private	94.4
[12]	Private	98.27
[8]	Private	94.5
Ours	Private	99

## Data Availability

The data presented in this study are available on request from the corresponding author. The data are not publicly available due to privacy.

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
