# Peer review of "A New Framework for Smart Doors Using mmWave Radar and Camera-Based Face Detection and Recognition Techniques"

_sensors, 2023, doi:10.3390/s24010172_

Round 1

Reviewer 1 Report

Comments and Suggestions for Authors

Decision: major revision,

1.       The paper mentions the integration of mmWave radar and camera sensors for face detection and recognition. The authors need to provide more details, how the integration is achieved? Are there any challenges or considerations in synchronizing data from these different sources?

2.       The authors emphasize improving security and convenience in access control systems. Can the authors elaborate, how the proposed system enhances security and convenience compared to existing smart door systems? Are there specific scenarios or use cases where this improvement is more pronounced? I didn’t find any information regarding this!

3.       The authors mentioned experiments showing the usefulness of the framework for smart home security. What specific experiments were conducted, and what were the key findings? Were there any limitations or challenges encountered during the experimental phase? Not provided any specification.

4.       The authors discuss the growing market for facial recognition door locks, citing figures and key players. How do the proposed system and its integration of mmWave radar align with or differentiate from current market trends in smart door security systems?

5.       The introduction mentions the impact of the COVID-19 pandemic on the demand for touch-free options. How does the proposed system address or contribute to the specific challenges posed by the pandemic in the context of smart door security? Not provided!

6.       The system using only cameras for person detection, especially in scenarios where individuals overlap. The authors didn’t provide insights into the limitations faced by camera-based systems in handling overlapping people, and how mmWave radar addresses this issue?

7.       The authors introduced the concept of fusing mmWave radar with a camera for improved accuracy. How this fusion is implemented, and what advantages it brings in terms of detection accuracy and reliability? Need more convincing answer and add details about this.

8.       The manuscript, however, does not link well with recent literature on recognition appeared in relevant top-tier journals, e.g., the IEEE Intelligent Systems department on AAD-Net: Advanced end-to-end signal processing system for human emotion detection & recognition using attention-based deep echo state network". Also, new trends of AI for recognition “ARTriViT: Automatic Face Recognition System Using ViT-Based Siamese Neural Networks with a Triplet Loss” are missing it should be comprised.

9.       The authors use of mmWave radar for person detection. How does the system handle scenarios with multiple individuals in its field of view?

10.   The mapping of radar coordinates into camera coordinates is discussed. How it contributes to the overall accuracy of the system?

Comments on the Quality of English Language

minor chaneges required!

Reviewer 2 Report

Comments and Suggestions for Authors

The article focuses on presenting an innovative solution for smart doors that combines facial detection and recognition techniques based on mmWave radar and camera sensors. In my opinion, the topic is interesting and quite innovative. This article is correctly formulated and is characterized by proper logic of coherence. The article presents a number of tests to enable the reader to understand the method used. Experimental results prove that the proposed framework has good future prospects, provided that the author needs to improve the processing speed and add a voice recognition function. In my opinion, after the first, the article is primarily of an engineering rather than scientific nature, after the second, some of the drawings are poorly presented, but these comments ultimately do not lead to misunderstandings.

I have a few more comments for users:

1. I suggest adding more newer links, as there are a lot of articles on this topic,

2. I propose a more detailed discussion of the literature review in the introduction,

3. Correct the drawings, e.g.:

- figure 3 does not contain explanations for this figure in the caption,

- figure 9 is not well presented and not explained in the caption,

...

4. I propose that the authors expand their conclusions more.

Reviewer 3 Report

Comments and Suggestions for Authors

This article proposes a smart door detection and identification solution, which has very important practical significance. This solution uses the idea of ​​multi-sensor combination to fuse optical images and millimeter-wave radar data to obtain improved performance results. However, there are the following issues that need to be explained:

1. The overall idea of the article seems to be a patchwork of several parts of hardware, and the corresponding algorithm parts are all existing. The author does not explain or emphasize his innovative points.

2. The introduction of each part of the algorithm is relatively simple. It does not highlight why the algorithm was chosen. There is no further explanation of the fused data or the advantages of the selected algorithm in this scenario. It is recommended that the author supplement it.

3. In the simulation part, it is necessary to increase the comparison with other algorithms to highlight the advantages of this algorithm.

Comments on the Quality of English Language

The grammar and wording of this article need further improvement, and I hope the author can further improve it.

Reviewer 4 Report

Comments and Suggestions for Authors

See the attach.

Round 2

Reviewer 1 Report

Comments and Suggestions for Authors

The authors addressed my comments and suggestions successfully! Good Luck!

Reviewer 3 Report

Comments and Suggestions for Authors

The author responded to all my comments and agreed to publish